# Prevalence, Clinico-Bacteriological Profile, and Antibiotic Resistance of Symptomatic Urinary Tract Infections in Pregnant Women

**DOI:** 10.3390/antibiotics12010033

**Published:** 2022-12-25

**Authors:** Rajani Dube, Shatha Taher Salman Al-Zuheiri, Mariyam Syed, Lekshmi Harilal, Dean Allah Layth Zuhaira, Subhranshu Sekhar Kar

**Affiliations:** 1Department of Obstetrics and Gynecology, RAK Medical and Health Sciences University, Ras Al Khaimah 11172, United Arab Emirates; 2Department of Obstetrics and Gynecology, Abdullah Bin Omran Hospital for Obstetrics and Gynecology, Ras Al Khaimah 11172, United Arab Emirates; 3Department of Pediatrics and Neonatology, RAK Medical and Health Sciences University, Ras Al Khaimah 11172, United Arab Emirates

**Keywords:** urinary tract infection in pregnancy, symptomatic UTI, clinical profile of UTI, bacteriological profile of UTI, antimicrobial resistance, pregnancy outcome in UTI, pregnancy complications in UTI

## Abstract

Background: Urinary tract infection (UTI) is a common complication in pregnancy. The prevalence varies between countries. This research aims at estimating the prevalence, clinico-bacteriological profile, antibiotic resistance, and risk factor analysis of symptomatic UTI in pregnancy. Method: This is a prospective observational study conducted at the Abdullah Bin Omran Hospital, RAK, UAE, from March 2019 to February 2020. All pregnant women attending the antenatal clinic during this period were given a pre-validated questionnaire for the symptoms of UTI. In symptomatic patients, urine was sent for microscopy, culture, and sensitivity. Women were treated for UTI and were followed up for the rest of the pregnancy. Data analysis was performed by SPSS software version 24 using descriptive statistics and comparisons with significance at a *p*-value of <0.05. Results: The prevalence of symptomatic UTI was 17.9%. *E.coli* was the commonest isolate followed by Group B *streptococcus*. The commonest symptom reported was loin pain and the most common risk factor was diabetes. Women with risk factors are significantly more likely to have culture-positive UTIs. Most of the pathogens were sensitive to cefuroxime and benzyl penicillin. Risk of preterm labor was higher. Conclusions: Regular antenatal care and routine urine testing in all visits are recommended for early detection and treatment of UTI.

## 1. Introduction

Urinary tract infection (UTI) is a common bacterial infection during pregnancy. It includes acute cystitis, pyelonephritis, and asymptomatic bacteriuria. Both symptomatic and asymptomatic UTI are significant causes of adverse maternal and fetal outcomes [1,2]. Based on available evidence, all over the world, the prevalence of symptomatic UTIs in pregnant women has been around 4–47% [2,3,4,5,6,7,8,9]. Signs and symptoms of symptomatic UTI include dysuria, increased urinary frequency, urgency, suprapubic pain, flank pain, hematuria, and fever [10]. Cystitis is defined as the presence of positive urine culture and symptoms of lower UTI (dysuria, urinary frequency, hematuria, urinary urgency, or suprapubic tenderness), in absence of systemic symptoms. Pyelonephritis is defined with positive urine culture and systemic symptoms (fever, chills, flank pain, or back pain) [10]. Asymptomatic bacteriuria (ASB), on the other hand, is diagnosed in women with high-burden bacterial growth without UTI symptoms. High burden is defined as the growth of bacteriuria of >10^5^ colony-forming units (CFU) per mL of urine of a single uro-pathogen and intermediate growth refers to bacteriuria with >10^3^–10^5^ CFU/mL of a single uro-pathogen. 

The bacterial isolation rate by culture can vary in different settings and by different methods used [11,12,13]. As high as 25–30% of symptomatic women may have a negative urine culture when a high-burden bacterial count threshold is used [14,15,16]. Hence, there is an increased reliance on signs and symptoms to diagnose the condition. The typical urinary symptoms, e.g., dysuria, frequency, and urgency, are highly predictive of a UTI in non-pregnant females and often therapy is empirically started without performing a urine culture in women with symptoms of an uncomplicated UTI [14,17,18,19,20,21]. However, the usual complaints of increased frequency, nocturia, and suprapubic pressure are not particularly helpful in pregnancy, because most pregnant women experience these as a result of increased pressure from the growing uterus, expanding blood volume, increased glomerular filtration rate, and increased renal blood flow [22]. Therefore, it is generally agreed that UTI should be diagnosed primarily by urine culture [23].

UTI during pregnancy is a significant cause of perinatal and maternal morbidity and mortality [1]. The maternal complications associated with UTI include preterm labor (labor onset before 37 weeks of gestation), hypertensive disorders of pregnancy (such as pregnancy-induced hypertension and preeclampsia), anemia (hematocrit level less than 30 percent), and amnionitis [24]. Neonatal outcomes that are associated with UTI include sepsis and pneumonia (specifically, group B streptococcus infection). There is also an increased risk of low-birth-weight infants (weight less than 2500 g), and stillbirth in the babies born in mothers with UTI [12,24,25,26].

The organisms that cause UTIs in pregnant women are the same as those found in non-pregnant patients. *Escherichia coli (E. coli)* accounts for 80 to 90 percent of infections. Other organisms include *Proteus mirabilis, Klebsiella pneumoniae (K. pneumoniae),* Group B *streptococcus (GBS),* and *Staphylococcus saprophyticus*. However, less common organisms like *Enterococci, Gardnerella vaginalis*, and *Ureaplasma ureolyticum* can also cause UTI [24]. In clinical practice, treatment for UTI is empirically started until the specific organism is identified from a urinalysis. The antibiotics (in order of preference) are nitrofurantoin, trimethoprim, and cephalexin. Amoxicillin is not suitable as empiric therapy for acute cystitis but can be used if urine culture shows susceptibility [24,27].

Physiological changes during pregnancy such as ureteral dilatation and incomplete emptying of the bladder may contribute to the development and spread of UTIs [28]. Apart from the pregnant state, there are additional risk factors for UTI in pregnancy, such as a past history of UTI, advanced maternal age, low educational level, low socioeconomic status, smoking, unsatisfactory personal hygiene, anemia, multi parity, as well as diabetes and sickle cell disease [5,29,30,31]. ASB without treatment can lead to symptomatic UTI and pyelonephritis in almost a quarter of the patients during pregnancy, although a study from the Netherlands suggested a low rate of pyelonephritis among low-risk women with ASB and uncomplicated singleton pregnancies without diabetes mellitus or urinary tract abnormalities without treatment [32,33,34].

A recent study from Abu Dhabi, UAE, shows a prevalence of 15% with the most common isolated organism being *GBS* [35]. However, a study from Ajman and Sharjah showed a lower prevalence rate of ASB with the most common organism being *E. coli* [8,36]. Furthermore, the antibiogram of organisms are different at different emirates of UAE [8,27,35,36]. The prevalence and sensitivity pattern of antibiotics can differ between countries, between the areas in the same country, and can change at the same place over a period of time. Data on local scenario helps in the formulation of guidelines, antibiotic stewardship, as well as improvement of fetal and maternal outcomes. Hence, this study aims at estimating the prevalence, studying the clinico-bacteriological profile, risk factors, antibiotic susceptibility of organisms, and the outcome of symptomatic UTI in pregnancy in the government hospital for obstetrics and gynecology at Ras Al Khaimah, UAE.

## 2. Results

### 2.1. Prevalence

A total of 682 women gave written consent and filled up the questionnaire. The complete questionnaires were included, and urine samples were sent. The prevalence of UTI based on symptoms and pus cells in urine microscopy is 17.9%. Urine culture showed bacterial isolates in 53 women with symptoms (11.2%) and 62.3% of total UTIs were culture positive [Figure 1].

### 2.2. Demography and Risk Factors

The mean age was 29.67 ± 3.58 years (range: 19–45 years), with the majority being aged 30 years or younger (n = 29; 54.7%). Furthermore, 17 (32%) women were primigravida, and most had parity ≥1 (n = 36; 67.9%). The mean body mass index (BMI) was 31.78 ± 11.18 kg/m^2^; the majority were obese with a BMI of ≥30 (n = 34; 64.1%), while 15.09% (n = 8) had class 3 obesity. The educational status for all women was higher secondary or above, whereas all of them belonged to middle or higher socio-economic status. Most cases were detected in the second trimester. When compared with first and third trimesters, the difference was not statistically significant (*p* > 0.5 for both). Among the known risk factors, the most common was diabetes (23%), followed by a previous history of UTI (19%), and anemia (15%). A total of 26.4% of cases occurred in women without any known risk factors for UTI other than the pregnant state (n = 14) in the culture-positive group, whereas 68.7% of women did not have risk factors in the culture-negative group (n = 22). When compared, this is significant at *p* < 0.05 [Table 1]. 

The most common symptoms were pain in the lower abdomen and frequency of urination. There were no differences in symptoms for culture-positive or culture-negative groups [Table 2].

### 2.3. Bacteriological Profile and Antibiotic Sensitivity

The most common organism isolated was *E. coli* (27%), followed by *GBS* (25%) and *K. pneumoniae*. Other organisms isolated included *Pseudomonas, Bacteroides, Citrobacter, Gardnerella,* and *Enterobacter cloacae* [Table 3]. Antibiotic therapy was started for patients if they were symptomatic and showed pus cells by microscopy after the sample for culture and sensitivity was obtained. Antibiotics were started for all suspected cases of preterm labor. For those with previous known GBS carriage and in labor (term or preterm labor), the initial antibiotic was benzyl penicillin and clindamycin (if they are penicillin sensitive). For all others with symptoms of UTI, cefuroxime was the first-line antimicrobial agent. Intrapartum prophylaxis was given to those with GBS isolates. The antibiotic was modified according to the sensitivity after the culture reports were available [Table 3].

The commonest antibiotic used successfully was cefuroxime for *E. coli* and benzyl-penicillin for *GBS.* Both *E. coli* and *Klebsiella* were highly sensitive to nitrofurantoin, amikacin, gentamicin, cefepime, ceftazidime, meropenem, and piperacillin/ tazobactam. *Klebsiella* was highly resistant to ampicillin (100%). There were 1 extended-spectrum beta-lactamase (ESBL)-producing *E. coli* and 1 ESBL-producing *Klebsiella* isolated, which were successfully treated with meropenem and nitrofurantoin [Table 4]. 

### 2.4. Clinical Profile and Pregnancy Outcome

The most common symptoms were pain in the lower abdomen (85%), dysuria (48%), and fever (48%). Lower abdominal pain and urinary frequency were present in a significant number of women and were not specific for either culture-positive or -negative UTIs. Among these, only 6.7% had culture-positive UTIs. Dysuria, painful micturition, fever, and urgency were more specific symptoms for UTI (>0.97). Hematuria was not reported by any participant. There was no difference between culture-positive or culture-negative UTIs with regard to any of the reported symptoms [Table 3]. The indication for hospitalization was suspected preterm labor or pyelonephritis. The most common organism isolated among women with pyelonephritis was *E. coli*, whereas GBS was most common among women with preterm labor. The prevalence of preterm labor was 33% (n = 28) in total. It was more common in women with culture-positive UTI [not significant (*p* = 0.071)] as compared with those with a culture-negative report. UTI was significantly more common in women over 30 years of age with preterm labor as compared with younger women (*p* = 0.03). When the outcomes were studied, early pregnancy loss was seen in six women (7%), and there was no increased risk of operative delivery in women with or without UTI. The mean hospital stay was 1.62 ± 1.3 days. Two women had a recurrence of UTI, and it was by *E. coli* (14.2%). Recurrence was not seen with any other organism. Pyelonephritis occurred in six women on further follow up and they were treated by piperacillin and tazobactam. Eleven of the women with UTI caused by GBS (out of 13) further received intrapartum prophylaxis with benzyl penicillin, while two had elective caesarean delivery. No cases of neonatal sepsis were found.

## 3. Discussion

UTI is a common infection in pregnant women in UAE. The prevalence is higher in our study, which may be due to the inclusion only of symptomatic women [3,5]. Approximately 37.7% of symptomatic women in our study did not show any growth of uro-pathogens in urine, which is more than some of the other studies [14,15,16]. This may be due to the fact that significant growth in this study is taken as a high-burden bacterial count (10^5^ CFU/mL). When explored further, none of the women had received any antibiotics in the past 2 weeks. 

The majority of women were UAE nationals and aged below 30 years. This is in accordance with another study from Abudhabi where UTI was more common in Emirati women [35]. In this study, we included 682 women to enquire about the symptoms of UTI. A total of 69.3% (n = 473) reported one or more symptoms of UTI. This was in contrast to previous studies in Saudi Arabia (12% symptomatic vs 8% asymptomatic) and Bangladesh (4.4% symptomatic vs 4.5% asymptomatic) [5,37]. In a recent study from Abudhabi, only 21% of women with UTI had one or more symptoms [35]. This can be explained by the fact that only women who had some symptoms at the presentation would have opted to be a part of the study. The association of demographic factors with UTI is not clear. Some studies show nulliparity and younger age to be associated with a higher prevalence of UTI [35,38]. Other studies show no significant associations of age, BMI, nationality, or parity [35,38,39]. In our study, most of the symptomatic women were in the secnd trimester, in contrast to another study by Balachandran et al. where they commonly presented in the first trimester [35].

The risk factors in our study were diabetes and previous history of UTI. While the previous history of UTI is a risk factor reported in many of the studies, diabetes is a risk factor for all infections [3,5,9,29,30,31,35]. However, unlike the previous reports, hypothyroidism was only present in two women with symptomatic UTI in this study [35,40]. Therefore, further studies are required in this area to find stronger associations. When compared with the culture-negative group, the culture-positive group had significantly more known risk factors (*p* = 0.0001). However, individual comparisons could not be performed due to the small numbers in the groups.

The most common pathogen isolated in this study was *E. coli,* but the prevalence was low (27%) compared with other studies [3,5,9,41]. The prevalence of *GBS* isolates (25%) and *K. pneumoniae* (20.7%) was higher compared with other studies [3,5,9,36,42,43]. There are no isolates of *staphylococcus*, unlike other studies [41]. This is in accordance with another study from UAE where the bacteria isolated were *GBS* (31.3%) and *E. coli* (30.9%), followed by *Enterococcus and K. pneumoniae* (13.1%) [35]. This local microbial population of the country may reflect geographical and sociocultural factors. *E. coli* being responsible for all recurrent UTIs in our study is in accordance with another study and could be explained by the increased susceptibility of uroepithelium to colonization by coliforms.

Antimicrobial resistance of *E. coli* and *GBS* was lower in this study compared with other countries [3,5,9]. *GBS* was quite sensitive to benzylpenicillin (92%), ampicillin (77%), and vancomycin (100%). This is in accordance with other studies where these antimicrobials were successfully used and vancomycin is preferred over clindamycin in penicillin-sensitive patients [35,38,44]. In the recent National Guidelines on Empiric Antibiotic Treatment of UTIs in Pregnant Women by the Ministry of Health and Prevention, it is recommended that nitrofurantoin should be used as the first line of treatment and amoxicillin/ clavulanic acid as an alternative [27]. Nitrofurantoin is avoided in women with glucose-6-phosphate dehydrogenase deficiency or those who are at higher risk for the condition. It is also best avoided in the third trimester due to concerns of its effect on fetal erythrocytes. It has not been associated with any birth defects in newborns. Antibiotics like penicillins, erythromycin, and cephalosporins are safe in pregnancy [45]. Our study also shows the high sensitivity of organisms to both of them. In addition, it also shows moderate sensitivity to cefuroxime (71%). Cefuroxime was the most common antibiotic used successfully as empirical therapy in culture-negative patients. 

Our study reported a higher rate of preterm delivery despite adequate treatment and a non-significant association of preterm labor with culture-positive UTIs. Preterm delivery was similarly reported as more common in other studies [32,35,46]. A recent Cochrane review also quotes a reduction in preterm labor with antibiotic treatment of UTI, but it was based on low-certainty evidence [32]. However, the culture-positive status was not significant, and the rate of preterm delivery remained high despite adequate treatment, in contrast to another study [47]. The maternal and fetal outcomes were favorable with no increase in low-birth-weight babies or neonatal sepsis. 

Screening and treatment of asymptomatic bacteriuria reduce pyelonephritis and perinatal complications. Though using symptom-based treatment can reduce complications, many women remain asymptomatic. Therefore, routine screening of pregnant women is recommended [27,33,48,49,50,51]. As there are concerns for antibiotic overuse and resistance, asymptomatic bacteriuria should be treated only on the basis of culture and sensitivity [50,52].

The limitation of this study is the non-inclusion of cases with asymptomatic bacteriuria. However, it is a prospective study and provides the microbiological status, antibiotic resistance, and pregnancy outcome, as well as a working questionnaire on symptoms of UTI in pregnancy.

## 4. Materials and Methods

This was a prospective observational study conducted at the Department of Obstetrics and Gynecology, Abdullah Bin Omran Hospital, Ras Al Khaimah, UAE, between 1 March 2019 and 29 February 2020. The ethical approval was obtained from the Ministry of Health and Prevention Research Ethics committee [MOHAP/REC/2019/30-2019-UG-M].

### 4.1. Sample Size and Sampling Technique

This study was performed on women attending the antenatal clinic of the Abdullah Bin Omran Hospital, Ras Al Khaimah, UAE. The sample size was determined using an online sample size calculator available at calculator.net and was found to be 341 based on our target population. As this was a prospective study, and dropouts were likely, all the women meeting the criteria were approached for the study by consecutive sampling until a total of 682 women consented, were enrolled, and filled in the questionnaire. Only women who were unable to understand and write English (n = 9) were excluded from the study. A validated questionnaire in English was then used to ask about the symptoms of UTI. For those who reported any symptoms, midstream clean-catch urine was sent for microscopy, culture, and sensitivity of urine. In addition, urine microscopy and culture were carried out for all women admitted with symptoms of UTI. 

The data collection sheet included socio-demographic details such as age, educational status, nationality, clinical characteristics such as body mass index (BMI), obstetric history, and the presence of known risk factors for UTI. The culture reports were studied for whether they were culture positive, for the type of organism, and the antibiotic sensitivity pattern. All pregnant women with culture-positive UTIs were followed prospectively for the rest of the pregnancy for studying the outcome. 

### 4.2. Questionnaire

We followed an existing guideline for developing the questionnaire [53]. The initial version of the questionnaire was prepared from available pre-validated questionnaires on uncomplicated UTIs [54,55]. It contained 12 items (questionnaire version 1) including the risk factors and known pregnancy complications. We invited faculty members (N = 10) from the institution to screen the questionnaire for content. We then invited non-pregnant student volunteers in the reproductive age group to test the questionnaire for general readability and comprehension. The final version of the questionnaire with only five questions and subheadings after the pilot testing was used for this study (questionnaire version 2) [Appendix A]. We tested the overall reliability of question one, with a score of six as the cut off, and the Cronbach α obtained was 0.76, suggestive of acceptable internal consistency.

### 4.3. Specimen Collection and Isolation

Women were instructed on the method of collection of midstream clean-catch urine. Microscopic urinalysis and urine cultures were used for the detection of UTI. The treatment was initiated. All samples were sent for culture and sensitivity before initiation of treatment. Blood agar and CLED media were used for culture. The VITEK 2 system (bioMérieux) was used for species-level identification.

### 4.4. Susceptibility Testing

The antibacterial susceptibility of all isolates was tested by automated susceptibility testing using Vitek 2 AST 03 card (BioMérieux, Inc., Durham, NC) according to the manufacturer's instructions with appropriate quality control strains, following Clinical Laboratory Standard Institute (CLSI, 2017) guidance [56]. The antibacterial agents tested included benzylpenicillin, ampicillin, amoxicillin, ceftriaxone, cefotaxime, ceftazidime, cefepime, clindamycin, gentamicin, nitrofurantoin, erythromycin, ofloxacin, levofloxacin, trimethoprim/ sulfamethoxazole, piperacillin, vancomycin, and linezolid. ESBL was identified when isolates were found to be resistant to cefepime, and ceftazidime, at MIC dilution of ≥64. Further genetic testing was not performed. 

### 4.5. Follow Up

Antibiotic therapy was started for patients after the sample for culture and sensitivity was obtained with cefuroxime. The antibiotics were modified according to the sensitivity report if required. A test of cure was performed for all culture-positive UTI cases 1 week after starting the antibiotic/ change of antibiotics. For the purpose of current research, recurrent UTI was defined as ≥2 infections in six months during pregnancy. Recurrences can represent reinfection or relapse. Persistent infection is defined as significant growth without an intervening negative report despite adequate therapy [28,57]. All women with culture-positive UTIs were followed prospectively for the rest of the pregnancy for studying the outcomes. 

### 4.6. Outcomes

The primary outcomes of this research include prevalence, bacteriological profile, and antibiogram. In addition, we studied symptom profiles likely to suggest UTI in pregnancy and compared the characteristics of groups with culture-positive and culture-negative UTIs. The pregnancy outcomes such as miscarriage, preterm labor, mode of delivery, length of hospital stay, and neonatal outcomes were recorded.

### 4.7. Data Analysis

Data analysis was performed using SPSS software version 24. Descriptive statistics, including percentages, mean, range, and standard deviations, were calculated for all variables. Simple descriptive statistics were used to describe the distribution of uro-pathogens and antibiotic sensitivity patterns. Proportions were compared using Chi-square tests, and a *p*-value less than 0.05 was considered statistically significant.

## 5. Conclusions

The prevalence of UTI is high in pregnant women attending Abdullah Bin Omran Hospital, Ras Al Khaimah, UAE. Symptoms of dysuria, fever, and urgency are predictors of UTI. The risk of preterm labor is high despite adequate treatment. Symptomatic women with risk factors are significantly more likely to have culture-positive than culture-negative UTIs. Routine screening for asymptomatic bacteriuria in early pregnancy is effective in preventing the occurrence of symptomatic UTIs and complications in pregnancy.

## Figures and Tables

**Figure 1 antibiotics-12-00033-f001:**
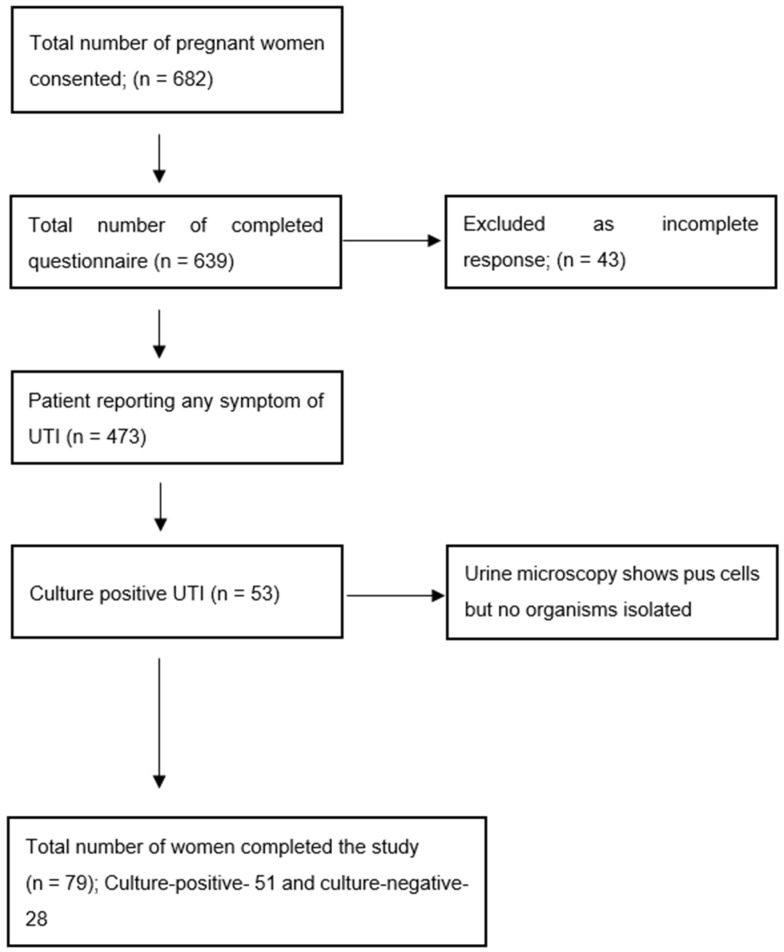
Patient inclusion.

**Table 1 antibiotics-12-00033-t001:** Comparison of demographic status of women with culture-positive and culture-negative UTI.

Parameter	Findings	Culture Positive (n = 53) (%)	Culture Negative (n = 32) (%)	Comparison
Age of the patient	≤30 years	29 (54.7)	18 (56.2)	*p* = 0.890 Not significant
	>30 years	24 (45.2)	14 (43.7)	
Nationality	Emirati	30 (56.6)	20 (62.5)	*p* = 0.592 Not significant
	Expatriate	23 (43.3)	12 (37.5)	
BMI (kg/m^2^)	<25	7 (13.2)	5 (15.6)	*p* > 0.05 Not significant
	25–29.9	12 (22.6)	7 (21.8)	
	30–34.9	19 (35.8)	12 (37.5)	
	35–39.9	7 (13.2)	4 (12.5)	
	≥40	8 (15%)	4 (12.5)	
Parity index	Primigravida	17(32)	9 (28.1)	*p* = 0.701 Not significant
	Parity 1 or above	36 (68)	23 (71.8)	
GA at presentation	First trimester (≤13 weeks)	11 (20.7)	7 (21.8)	
	Second trimester (13^+1^–26^+6^ weeks)	24 (45.2)	15 (46.8)	
	Third trimester (≥27^+0^ weeks)	18 (34)	10 (31.2)	
Risk factors	Diabetes ^1^	12 (22.6)	5 (15.6)	Presence of any risk factors— *p* = 0.0001 significant
	Previous history of UTI	10 (19)	1 (3.12)	
	Anemia ^2^	8 (15)	2 (6.25)	
	Hypertension ^3^	4 (7.5)	1(3.12)	
	Others ^4^	5 (9.4)	1(3.12)	

^1^—Gestational diabetes (n = 9) and pre-gestational diabetes, (n = 3); ^2^—Included 14 cases of iron deficiency anemia and 1 sickle cell anemia; ^3^—Hypertension is defined as gestational hypertension + preeclampsia; ^4^—Female genital mutilation (n = 1), hypothyroidism (n = 2), recent catheterization (n = 1), multiple pregnancy (n = 1).

**Table 2 antibiotics-12-00033-t002:** Symptoms and correlation with Culture positive UTI (n = 473).

Symptoms (n)	Culture Positive (n = 53) (True Positive)	Culture-Negative UTI (Treated) (n = 32)	No UTI (n = 388)	OR * for Symptoms and Culture Positivity (95%CI)
Pain in the lower abdomen (n = 423)	45 (84.9%)	27 (84.3%)	351	OR = 1.041(0.309–3.511) *p* = 0.473
Burning during urination (n = 61)	26 (49%)	15 (46.8%)	20	OR = 1.091 (0.453–2.628) *p* = 0.422
Pain during urination (n = 23)	11 (20.7%)	7 (21.8%)	5	OR = 0.935 (0.321–2.725) *p* = 0.451
Frequency of urination (n = 355)	24 (45.2%)	14 (43.7%)	317	OR = 1.064 (0.440–2.574) *p* = 0.445
Urinary urgency (n = 49)	24 (45.2%)	14 (43.7%)	11	OR = 1.064 (0.440–2.574) *p* = 0.445
Fever with/without chills (n = 47)	26 (49%)	15 (46.8%)	6	OR = 1.091 (0.453–2.628) *p* = 0.422
Urinary incontinence (n = 19)	7 (13.2%)	3 (9.37%)	9	OR = 1.471 (0.352–6.148) *p* = 0.528
Urinary retention (n = 3)	2 (3.77%)	0	1	

* OR = Odds Ratio.

**Table 3 antibiotics-12-00033-t003:** Culture isolates and antibiotic sensitivity.

Organism Isolated (n)	Initial Antibiotic (n)	Second-Line Antibiotic (n)	Comment
*E. coli* (14)	Cefuroxime (14)	Meropenem + nitrofurantoin (1) Piperacillin + tazobactam (3)	-ESBL strain treated with meropenem and NFT (n = 1) -Pyelonephritis treated with piperacillin+ tazobactam (n = 3)
GBS (13)	Cefuroxime (7) BZP (4) Clindamycin (2)	BZP (10) Piperacillin + tazobactam (1) Vancomycin (2)	-Cases with known penicillin sensitivity were treated with vancomycin (n = 2) later as they showed clindamycin resistance -Pyelonephritis treated with piperacillin+ tazobactam (n = 1)
*Klebsiella* (11)	Cefuroxime (11)	Piperacillin + tazobactam (2) Gentamicin (1) NFT (1)	-ESBL strain treated with meropenem and NFT (n = 1) -Pyelonephritis treated with piperacillin+ tazobactam (n = 2)
*Bacteroides* (5)	Cefuroxime (5)	Metronidazole (4) Clindamycin (1)	
*Gardnerella* (2) *Pseudomonas* (3) *Proteus* (2) *Citrobacter* (2) *Enterobacteria cloacae* (1)	Cefuroxime (10)	Meropenem, cephazolin, Azithromycin (1) Nitrofurantoin (1) Piperacillin + tazobactam (3)	-Meropenem, cephazolin, azithromycin was used for enterobacteria cloacae -Pseudomonas treated with piperacillin+ tazobactam (n = 3)
No growth (32)	Cefuroxime (32)		

Nitrofurantoin = NFT; Benzyl-penicillin = BZP.

**Table 4 antibiotics-12-00033-t004:** Culture isolates and antibiotic sensitivity.

Bacteria	*Escherichia coli* (n = 14) (Sensitivity %)	Group B *Streptococcus* (n = 13) (Sensitivity %)	*Klebsiella pneumoniae* (n = 11) (Sensitivity %)
Antibiotic sensitivity	Nitrofurantoin (n = 13, 92.8) Cefuroxime (n = 10, 71.4) Cefotaxime (n = 4, 28.5) Ceftriaxone (n = 9, 64.2) Ceftazidime (n = 10, 71.4) Cefepime (n = 11, 78.5) Gentamicin (n = 12, 85.7) Amikacin (n = 13, 92.8) Aztreonam (n = 7, 50) Meropenem (n = 13, 92.8) Ampicillin (n = 4, 28.5) Amoxicillin/Clavulanic acid (n = 10, 71.4) Piperacillin/Tazobactam (n = 13, 92.8) Ciprofloxacin (n = 3, 21.4)	Vancomycin (n = 13, 100) Linezolid (n = 12, 92.3) Ceftriaxone (n = 12, 92.3) Cefotaxime (n = 10, 76.9) Ampicillin (n = 10, 76.9) Benzylpenicillin (n = 12, 92.3) Levofloxacin (n = 11, 84.6) Erythromycin (n = 6, 46.1) Clindamycin (n = 6, 46.1) Trimethoprim/Sulfamethoxazole (n = 4, 30.7)	Nitrofurantoin (n = 10, 90.9) Cefuroxime (n = 7, 63.6) Cefotaxime (n = 7, 63.6) Ceftriaxone (n = 7, 63.6) Ceftazidime (n = 8, 72.7) Cefepime (n = 10, 90.9) Gentamicin (n = 10, 90.9) Amikacin (n = 10, 90.9) Aztreonam (n= 6, 54.5) Meropenem (n = 10, 90.9) Ampicillin (n = 4, 36.3) Amoxicillin/Clavulanic acid (n = 7, 63.6) Piperacillin/Tazobactam (n = 10, 90.9) Ciprofloxacin (n = 3, 27.2)

## Data Availability

The raw data are available from the corresponding author and can be provided on reasonable request. They have not been deposited in any repository.

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
