# Peer review of "Prevalence, Clinico-Bacteriological Profile, and Antibiotic Resistance of Symptomatic Urinary Tract Infections in Pregnant Women"

_antibiotics, 2022, doi:10.3390/antibiotics12010033_

Round 1
Reviewer 1 Report
The article is interesting, but the editing and presentation of the results require multiple improvements.
Furthermore
1. Chart -2 does not bring new information, it should be improved
2. Medical terms are necessary to be used instead of "pain during urination", and "visible blood in the urine".
Author Response
Dear reviewer,
Thank you for your time and effort.
The entire manuscript is thoroughly revised and modified.
1. Chart -2 does not bring new information, it should be improved |
As per suggestion Chart-2 is removed from the manuscript |
2. Medical terms are necessary to be used instead of "pain during urination", and "visible blood in the urine". |
Suggested changes incorporated and highlighted yellow |
With warm regards,
Reviewer 2 Report
The manuscript entitled " Prevalence, Clinico-bacteriological profile, antibiotic resistance of symptomatic Urinary Tract Infections in pregnant women " in which the authors focused on the prevalence of the urinary tract infections (UTI) in pregnant women. They also demonstrated the importance of regular antenatal care and routine urine testing in all visits of pregnant women for early detection and treatment of UTI.
The work is understandable and the topic is important. The scientific narrative is well structured and flows naturally from one idea to the next.
However, this paper suffers from few shortcomings that if modified would make the manuscript very suitable for publication in Antibiotics Journal.
Shortcomings:
1- The authors write in conclusion part line 329“The prevalence of UTI is high in UAE”. I think if the authors write “The prevalence of UTI is high in pregnant women attending Abdullah Bin Omran Hospital, Ras Al Khaimah, UAE is high” is better because the authors didn’t study many different hospitals in UAE.
2- I am wondering why the author use the English questionnaire only and didn’t use also the Arabic translated copy of the English questionnaire to include more pregnant women in their study who didn’t understand the English because the authors mentioned that “Only women who were unable to understand and write English were excluded from the study” in line 276.
3- The scientific narrative is well structured and understandable. However, the editing like font and putting distance between words needs significant improvement. Please revise the whole paper for the editing.
Below is some advice to change (related to typos and language):
· “Commonest symptom” in line 27. Please add The commonest symptom.
· “It includes acute cystitis, pyelonephritis and asymptomatic bacteriuria. “Please adjust the font of sentences in introduction, lines 37, 38.
· Please remove “FINAL REFERENCES” in line 353.
· The authors used “kg/mt2” for body mass index but “mt” isn’t common abbreviation for meter. I think it is better to be “m”.
· Please remove” urgency” because it is repeated. “urgency, and urgency were more specific symptoms for UTI (>0.97)” in line 181.
· “was Diabetes(23%), followed by a previous history of UTI (19%), and anemia (15%)………..in line 147. Please write diabetes with small letter and revise the whole manuscript.
· Please put space after the end of sentence and beginning of sentence and also between words in abstract part and in the whole manuscript.
For example
. Results:The prevalence of symptomatic UTI was 17.9%.E.coli was 26
the commonest isolate followed by Group B streptococcus. Commonest symptom reported was loin 27
pain and the most common risk factor was Diabetes.Women with risk factors are significantly more 28
likely to have culture-positive UTIs.Most of the pathogens were sensitive to Cefuroxime and Ben- 29
zylPenicillin.Risk of preterm labor. Conclusions:Regular antenatal car
Author Response
Dear reviewer,
Thank you for your time and efforts.
The entire manuscript is thoroughly revised and modified. All the suggested changes were incorporated and highlighted.
The entire manuscript is thoroughly revised and modified.
Comments and Suggestions for Authors
The manuscript entitled " Prevalence, Clinico-bacteriological profile, antibiotic resistance of symptomatic Urinary Tract Infections in pregnant women " in which the authors focused on the prevalence of the urinary tract infections (UTI) in pregnant women. They also demonstrated the importance of regular antenatal care and routine urine testing in all visits of pregnant women for early detection and treatment of UTI.
The work is understandable and the topic is important. The scientific narrative is well structured and flows naturally from one idea to the next.
However, this paper suffers from few shortcomings that if modified would make the manuscript very suitable for publication in Antibiotics Journal.
Shortcomings:
1- The authors write in conclusion part line 329“The prevalence of UTI is high in UAE”. I think if the authors write “The prevalence of UTI is high in pregnant women attending Abdullah Bin Omran Hospital, Ras Al Khaimah, UAE is high” is better because the authors didn’t study many different hospitals in UAE. DONE
2- I am wondering why the author use the English questionnaire only and didn’t use also the Arabic translated copy of the English questionnaire to include more pregnant women in their study who didn’t understand the English because the authors mentioned that “Only women who were unable to understand and write English were excluded from the study” in line 276.
The reason for excluding the women who did not understand English is that almost all the women in our study were educated and well-versed in English. There were only 9 women who could not sufficiently understand English till our desired number of samples was reached. Furthermore, at the time of the conduction of the study, a validated Arabic version for use in pregnant women was not available. An Arabic translation was not well received. However, further studies will include the Arabic-speaking population.
3- The scientific narrative is well structured and understandable. However, the editing like font and putting distance between words needs significant improvement. Please revise the whole paper for the editing. The entire manuscript is thoroughly revised and modified.
Below is some advice to change (related to typos and language):
- “Commonest symptom” in line 27. Please add The commonest symptom. DONE
- “It includes acute cystitis, pyelonephritis and asymptomatic bacteriuria. “Please adjust the font of sentences in introduction, lines 37, 38. DONE
- Please remove “FINAL REFERENCES” in line 353. DONE
- The authors used “kg/mt2” for body mass index but “mt” isn’t common abbreviation for meter. I think it is better to be “m”. DONE
- Please remove” urgency” because it is repeated. “urgency, and urgency were more specific symptoms for UTI (>0.97)” in line 181. DONE
- “was Diabetes(23%), followed by a previous history of UTI (19%), and anemia (15%)………..in line 147. Please write diabetes with small letter and revise the whole manuscript. DONE
- Please put space after the end of sentence and beginning of sentence and also between words in abstract part and in the whole manuscript.
This was done only in the abstract section, as the number of words were limited to 200. However, this has been modified.
For example
. Results:The prevalence of symptomatic UTI was 17.9%.E.coli was 26
the commonest isolate followed by Group B streptococcus. Commonest symptom reported was loin 27
pain and the most common risk factor was Diabetes.Women with risk factors are significantly more 28
likely to have culture-positive UTIs.Most of the pathogens were sensitive to Cefuroxime and Ben- 29
zylPenicillin.Risk of preterm labor. Conclusions:Regular antenatal car
Reviewer 3 Report
The manuscript titled “Prevalence, Clinico-bacteriological profile, antibiotic resistance of symptomatic Urinary Tract Infections in pregnant women” can be considered for publication after these major changes.
1. A minor change in title is suggested “Prevalence, Clinico-bacteriological profile and antibiotic resistance of symptomatic Urinary Tract Infections in pregnant women”
2. The Materials and Methods section is not in sequence. The authors can include ethical review followed by patients, Questionnaire, Specimen collection, isolation. Identification susceptibility testing, follow up for cure (that should include the antibiotics used), Outcomes and Data analysis. Please reformat the methodology.
3. The spelling of Gentamicin need to be correct throughout the manuscript.
4. The antibacterial susceptibility of all isolates was tested by automated susceptibility testing using Vitek 2, the data for the MIC can be included as well.
5. The Authors have mentioned that “Nitrofurantoin should be used as the first line of treatment and Amoxycillin/clavulanic acid as an alternative [27]. Those are safe in pregnancy. However, Nitrofurantoin may be taken during pregnancy, but it is generally best avoided in the third trimester because there's a small chance it could cause problems with your baby's red blood cells. Please discuss.
6. The authors have mentioned “There were 1 extended-spectrum beta-lactamase (ESBL)-producing E. coli and 1 ESBL-producing Klebsiella isolated” In methodology no such information is provided. How were the isolates checked for ESBL production? Phenotypically? Genotypically?
7. The antibiotics were also used to treat the patients. It is not clear whether the treatment was done in response to the antimicrobial testing, or the patients were started empirical therapies. Please make this portion clarify and provide sufficient details to understand. How many patients were treated using What antimicrobial agents.
Author Response
Dear reviewer,
Thank you for your time and efforts.
The entire manuscript is thoroughly revised and modified.
Comments and Suggestions for Authors
The manuscript titled “Prevalence, Clinico-bacteriological profile, antibiotic resistance of symptomatic Urinary Tract Infections in pregnant women” can be considered for publication after these major changes.
- A minor change in title is suggested “Prevalence, Clinico-bacteriological profile and antibiotic resistance of symptomatic Urinary Tract Infections in pregnant women” DONE
- The Materials and Methods section is not in sequence. The authors can include ethical review followed by patients, Questionnaire, Specimen collection, isolation. Identification susceptibility testing, follow up for cure (that should include the antibiotics used), Outcomes and Data analysis. Please reformat the methodology. DONE
- The spelling of Gentamicin need to be correct throughout the manuscript. DONE
- The antibacterial susceptibility of all isolates was tested by automated susceptibility testing using Vitek 2, the data for the MIC can be included as well. DONE
- The Authors have mentioned that “Nitrofurantoin should be used as the first line of treatment and Amoxycillin/clavulanic acid as an alternative [27]. Those are safe in pregnancy. However, Nitrofurantoin may be taken during pregnancy, but it is generally best avoided in the third trimester because there's a small chance it could cause problems with your baby's red blood cells. Please discuss. DONE
- The authors have mentioned “There were 1 extended-spectrum beta-lactamase (ESBL)-producing E. coli and 1 ESBL-producing Klebsiella isolated” In methodology no such information is provided. How were the isolates checked for ESBL production? Phenotypically? Genotypically? DONE
The VITEK 2 system (bioMérieux) was used for species-level identification. The isolates were tested for sensitivity in automated strips according to the manufacturer's instructions with appropriate quality control strains. ESBL was identified when isolates were found to be resistant to cefepime, and ceftazidime, at MIC dilution of ≥ 64. Further genetic testing was not performed.
- The antibiotics were also used to treat the patients. It is not clear whether the treatment was done in response to the antimicrobial testing, or the patients were started empirical therapies. Please make this portion clarify and provide sufficient details to understand. How many patients were treated using What antimicrobial agents.
SECTION MODIFIED AND CLARIFIED-DONE. Table 3 was incorporated to include detailed information
Round 2
Reviewer 3 Report
The authors have improved the manuscript. Please look into the spelling and typographical mistakes.